# Hypothyroidism Is a Predictive Factor for Better Clinical Outcomes in Patients with Advanced Hepatocellular Carcinoma Undergoing Lenvatinib Therapy

**DOI:** 10.3390/cancers12113078

**Published:** 2020-10-22

**Authors:** Masako Shomura, Haruka Okabe, Emi Sato, Kota Fukai, Koichi Shiraishi, Shunji Hirose, Kota Tsuruya, Yoshitaka Arase, Kazuya Anzai, Tatehiro Kagawa

**Affiliations:** 1Faculty of Nursing, Tokai University School of Medicine, 143 Shimokasuya Isehara City, Kanagawa 2591193, Japan; hokabe@tsc.u-tokai.ac.jp (H.O.); se111809@tsc.u-tokai.ac.jp (E.S.); 2Department of Preventive Medicine, Tokai University School of Medicine, 143 Shimokasuya Isehara City, Kanagawa 2591193, Japan; kota229@tokai.ac.jp; 3Division of Gastroenterology, Department of Internal Medicine, Tokai University School of Medicine, 143 Shimokasuya Isehara City, Kanagawa 2591193, Japan; shiraishik@msj.biglobe.ne.jp (K.S.); hs099212@tsc.u-tokai.ac.jp (S.H.); ktsuruya@tokai-u.jp (K.T.); ay114535@tsc.u-tokai.ac.jp (Y.A.); k_anzai@tokai-u.jp (K.A.); kagawa@tokai.ac.jp (T.K.)

**Keywords:** adverse events, hepatocellular carcinoma, hypothyroidism, lenvatinib

## Abstract

**Simple Summary:**

Patients with advanced hepatocellular carcinoma (HCC) undergoing molecular targeted therapy often experience non-negligible adverse events (AEs). Paradoxically, certain AEs are reportedly associated with a good prognosis. We aimed to identify factors predictive of treatment duration and overall survival (OS) in patients with HCC undergoing lenvatinib therapy. This study suggested that better baseline liver function was predictive of longer treatment duration and better prognosis in patients with advanced HCC treated with lenvatinib. Moreover, an AE of Grade 2/3 hypothyroidism was associated with a better prognosis in patients receiving lenvatinib treatment for advanced HCC. Continuing anticancer therapy with appropriate thyroid hormone replacement may contribute to longer survival.

**Abstract:**

Patients with advanced hepatocellular carcinoma (HCC) undergoing molecular targeted therapy often experience non-negligible adverse events (AEs). Paradoxically, certain AEs are reportedly associated with a good prognosis. We aimed to identify factors predictive of treatment duration and overall survival (OS) in patients with HCC undergoing lenvatinib therapy. Forty-six consecutive patients with advanced HCC who received lenvatinib therapy from April 2018 to November 2019 were prospectively followed until November 2019. Treatment efficacy was assessed according to the modified Response Evaluation Criteria in Solid Tumors for 2–3 months after therapy initiation. The disease control rate (DCR) was defined as the percentage of patients with a complete response, partial response, or stable disease. The DCR was 65.2%, with a median survival of 10.2 months. Grade 2/3 hypoalbuminemia resulted in shorter treatment duration. Factors predictive of longer OS were a Child-Pugh score of 5 at baseline and the occurrence of Grade 2/3 hypothyroidism. Conversely, Grade 2/3 hypoalbuminemia was associated with a poorer prognosis. An AE of Grade 2/3 hypothyroidism was associated with a better prognosis in patients receiving lenvatinib treatment for advanced HCC. Continuing anticancer therapy with appropriate thyroid hormone replacement may contribute to longer OS.

## 1. Introduction

Hepatocellular carcinoma (HCC) is the second leading cause of cancer-related death worldwide [1]. It is also an intractable cancer considering its high rate of recurrence even after curative therapies [2,3]. In particular, vascular invasion and extrahepatic metastasis greatly decrease survival rates [4].

Currently, sorafenib [5,6] and lenvatinib [7,8] are used as first-line therapies, and regorafenib [9] and ramucirumab [10] are used as second-line therapies for unresectable HCC in Japan. These therapies contribute to improving clinical outcomes in patients with advanced HCC.

Patients with advanced HCC undergoing molecular targeted therapy often experience non-negligible adverse events (AEs) that aggravate prognosis by necessitating a reduction in the scheduled doses or discontinuation of the treatment. Paradoxically, certain AEs are reportedly associated with a good prognosis. For example, patients treated with sorafenib who had a hand/foot skin reaction survived longer than those without this reaction [11]. In addition, any grade hypertension [12] or Grade 2/3 diarrhea [13,14] was associated with longer overall survival (OS).

Lenvatinib is an oral tyrosine kinase inhibitor (TKI) targeting vascular endothelial growth factor (VEGF) receptors 1–3, fibroblast growth factor receptors 1–4, RET, KIT, and platelet-derived growth factor receptor α [7,8]. Lenvatinib was approved as the first-line therapy against advanced HCC in 2018 in Japan. Several baseline variables, such as age, sex, performance status, and tumor stage, were associated with treatment outcomes [7,8]. Multiple factors may affect clinical/treatment outcomes of lenvatinib therapy for HCC. According to previous studies, preserved hepatic function (e.g., lower Child-Pugh scores or modified albumin and bilirubin [mALBI] grades) have a favorable relationship with OS [15]. The number of tumors and the size of the target lesion were associated with improved treatment efficacy [16]. Lower alpha-fetoprotein (AFP) values and early tumor shrinkage contributed to better progression-free survival (PFS) [16].

In lenvatinib therapy for thyroid cancer, AEs such as hypertension [17] and diarrhea [18] were related to a good prognosis. In the post hoc analysis of phase 3 REFLECT trial comparing the efficacy of lenvatinib with that of sorafenib, the occurrence of hypertension, diarrhea, proteinuria, or hypothyroidism was associated with longer OS in patients treated with lenvatinib [19]. However, information on the association of AE occurrence with prognosis in lenvatinib therapy for HCC is limited. Ohki et al. reported that lenvatinib-induced hypothyroidism of Grade ≥2 was an independent predictor of poorer PFS [20], whereas Koizumi et al., found that thyroid dysfunction was associated with better PFS [21]. Therefore, the impact of thyroid-related AE on prognosis is controversial. In this study, we aimed to identify factors, with special attention to AEs, that were predictive of treatment duration and OS in patients with HCC undergoing lenvatinib therapy.

## 2. Results

### 2.1. Baseline Patient Characteristics

Overall, 46 patients were enrolled for 19 months. Baseline Patient Characteristics are shown in Table 1. Most patients were men (80%), were aged 75 years and older (52%), had a Child-Pugh score of 5 (63%), and had a Barcelona Clinic Liver Cancer Stage C (54%) [22]. Eighteen (39%) patients started lenvatinib at a reduced dose because poor tolerability was anticipated.

### 2.2. Treatment Efficacy

Treatment efficacy was evaluated in 46 patients, including two patients who died of disease progression before the first evaluation. Disease control was achieved in 30 patients (65.2%), including one (2.2%) with complete response (CR), 15 (32.6%) with partial response (PR), and 14 (30.4%) with stable disease (SD) at 2–3 months after the initiation of lenvatinib therapy. The median treatment duration and OS were 7.7 and 10.2 months, respectively.

### 2.3. Adverse Events

Adverse Events are shown in Table 2. All 46 patients experienced at least one AE. The most common AEs (all grades) during the observation period were fatigue (85%), followed by appetite loss (67%), proteinuria (50%), and weight loss (50%). Grade 3 AEs were observed in 21 patients (46%), with fatigue, anorexia, and proteinuria being most frequent (15%). We observed Grade 1, 2, and 3 hypothyroidism in 10 cases (22%), nine cases (20%), and one case (2%) in the first month, in 15 cases (36%), 11 cases (26%), and no case in the second month, and in 11 cases (28%), 11 cases (28%), and no patients in the third month, respectively. As a result, Grade 2 and 3 hypothyroidism occurred in 12 patients (26%) and one (2%) patient, respectively, during the observation period; however, this AE was well controlled by adjusting the levothyroxine doses according to serum thyroid-stimulating hormone (TSH) levels. No AE-related deaths were reported. Fourteen (30%) and 12 (26%) patients required transient or permanent discontinuation of lenvatinib therapy owing to AEs, respectively.

### 2.4. Factors Associated with Treatment Duration

According to the log-rank analysis, better mALBI grade and hypoalbuminemia grade may contribute to longer treatment duration (Figure 1A,B). However, Cox analysis and sensitivity analysis showed that only Grade 2/3 hypoalbuminemia was significantly associated with a shorter treatment duration (hazard ratio (HR) (95% confidence interval, CI) was adjusted for age = 4.28 (1.28–15.00), Model 2 = 5.01 (1.37–18.35), and Model 4 = 5.65 (1.55–20.53) (Table 3).

### 2.5. Factors Associated with OS 

Among baseline variables, lower mALBI grade (Figure 2A) and a Child-Pugh score of 5 (Figure 2B) were associated with longer OS using log-rank tests. The association of AEs with OS was evaluated using landmark analysis at 3 months after lenvatinib therapy initiation. Interestingly, the patients who had Grade 2/3 hypothyroidism had significantly longer OS than those who had Grade 0/1 hypothyroidism according to the log-rank test and any models in the sensitivity analysis (age-adjusted HR = 0.21, 95% CI: 0.05–0.94, Model 1 = 0.20 (0.04–0.87), Model 2 = 0.19 (0.04–0.87), Model 3 = 0.18 (0.04–0.87), Model 4 = 0.16 (0.04–0.77), Figure 2C). The disease control rate (DCR) was 91% (10/11) in patients with Grade 2/3 hypothyroidism, which was higher than that in those with Grade 0/1 hypothyroidism (71% [20/28]); however, the difference was not significant. In contrast, Grade 2 hypoalbuminemia (age-adjusted HR = 4.96, 95% CI: 1.36–18.12, Model 2 = 5.83 (1.45–23.48), Model 4 = 5.83 (1.54–22.10), Figure 2D) and Grade 2 dysgeusia (age-adjusted HR = 3.55, 95% CI: 1.27–9.88, Figure 2E) were associated with a poor prognosis (Table 4).

## 3. Discussion

This study demonstrated that better baseline liver function was predictive of longer treatment duration and better prognosis in patients with advanced HCC treated with lenvatinib. Among lenvatinib-induced AEs, the occurrence of Grade 2/3 hypothyroidism was associated with longer OS, whereas Grade 2 hypoalbuminemia and dysgeusia were related to shorter OS.

A lower mALBI grade and Child-Pugh score contributed to longer treatment duration and longer OS in our study. The association of preserved liver function with a better prognosis has been repeatedly reported in the treatment of HCC [23]. Similar results were obtained with lenvatinib therapy; patients with baseline mALBI Grade 1 or a Child-Pugh score of 5 had a longer treatment duration [24,25], better treatment efficacy, and better outcomes [26]. In contrast, unexpectedly, tumor-related factors (e.g., tumor size, extrahepatic metastasis, vascular invasion, or serum AFP values) were not found to be associated with prognosis in this study, which is in agreement with a recent study demonstrating that hepatic reserve function (Child-Pugh score and mALBI grade) but not tumor burden (TNM stage, vascular invasion, and AFP) was predictive of survival in 152 patients who received lenvatinib [15]. Therefore, hepatic function might have a greater impact than tumor burden on outcomes of patients receiving lenvatinib therapy.

Lenvatinib-induced hypothyroidism occurred in 19 (41%) patients, with six (13%) patients having Grade 1, 12 (26%) having Grade 2, and one (2%) having Grade 3. This occurrence rate was higher than the 16–22% of patients with any grade hypothyroidism in phase 2 and 3 studies [7,8]. The reason for this difference is unclear; however, even higher occurrence rates, such as 66%, were reported in a recent real-world study [21]. We found that the presence of Grade 2/3 hypothyroidism at month 3 was a better prognostic marker in patients receiving lenvatinib therapy for advanced HCC. The association of lenvatinib-induced hypothyroidism with clinical outcomes is controversial. Koizumi et al. reported similar results [21] to ours, whereas Ohki et al., related this AE to poor survival [20]. Several studies analyzed the impact of therapy-induced hypothyroidism on prognosis. In the treatment of metastatic renal cell carcinoma, sunitinib-induced hypothyroidism, which had an incidence as high as 85% [27], was associated with longer PFS [28] and OS [29]. A retrospective study analyzing 538 patients who received TKI for advanced nonthyroidal cancers demonstrated that TKI-induced overt hypothyroidism (TSH ≥ 10 mIU/L, low free thyroxine, or requiring replacement) was significantly associated with longer OS (HR: 0.561) after adjusting for age, sex, race/ethnicity, cancer type, cancer stage, performance status, and checkpoint inhibitor therapy [30]. Hence, the hypothesis that patients who have hypothyroidism as an AE have a better prognosis than those who do not may apply to patients receiving lenvatinib therapy, as shown in our study.

The pathogenesis of TKI-induced hypothyroidism remains unknown. Supposed mechanisms include destructive thyroiditis, prevention of VEGF binding to normal thyroid cells and/or impairing thyroid blood flow, reduced synthesis of thyroid hormones, inhibition of iodine uptake, RET impairment, and progressive damage to thyroid function [31]. As most TKIs cause thyroid dysfunction, this AE could be a class effect of TKI [31]. Patients who progress to hypothyroidism might be more sensitive to TKI than those who do not. Such patients might manifest a greater response to this drug, which could result in better outcomes. Therefore, continuing TKI therapy with appropriate thyroid hormone replacement could be important in achieving longer survival.

The occurrence of treatment-related Grade 2 hypoalbuminemia at month 3 had a negative impact on treatment duration and OS in our study. There was no study focusing on hypoalbuminemia in patients receiving lenvatinib therapy. Our previous study revealed that Grade 2 hypoalbuminemia was associated with shorter OS in patients receiving sorafenib therapy for advanced HCC [11]. Thus, the serum albumin level, both at baseline and during therapy, is an important predictive marker to predict patients’ outcomes.

Lenvatinib-induced dysgeusia was also associated with a poorer prognosis. The proposed causes of dysgeusia include liver dysfunction, hypothyroidism, and oral mucositis [32]. In our study, the occurrence rate of Grade 2 dysgeusia was 11 cases (28%) at month 3. Moreover, the 11 patients with dysgeusia included three patients with Grade 2/3 oral mucositis, eight with Grade 1/2 hypothyroidism, and eight with a Child-Pugh score of 6 or above at month 3. The lenvatinib-induced dysgeusia may be due to hypothyroidism and liver dysfunction caused by the therapy. More studies are needed to assess these hypotheses.

Our study suggested that we could predict the clinical course based on the occurrence of AEs. Patient education and prophylactic medications should be provided at treatment initiation to appropriately manage AEs to achieve better treatment outcomes.

This study had limitations. We conducted the study in only one institution, and the sample size was relatively small. Because of the small number of patients, we analyzed factors influencing treatment duration and OS using Cox regression models adjusted only for age. Therefore, confounding among variables was unavoidable. However, we added sensitivity analysis and time-dependent Cox regression analysis, which could reduce the risk of cofounding in this study. A multicenter study with a larger number of patients is necessary to verify our results.

## 4. Materials and Methods

### 4.1. Ethics

This study was conducted according to the Declaration of Helsinki (2000) of the World Medical Association. Ethical approval was obtained from the Institutional Review Board of Tokai University Hospital (NO16R-023). All patients provided written informed consent.

### 4.2. Patients

This prospective study enrolled consecutive patients with advanced HCC who received lenvatinib therapy from April 2018 to November 2019. Patients completed surveys at treatment initiation (baseline) and every month. All patients were provided nursing intervention programs, such as education regarding self-monitoring and AE management, and telephone consultations.

### 4.3. Treatment Procedures

Patients received lenvatinib orally at a dose of 12 mg/day (for bodyweight ≥ 60 kg) or 8 mg/day (for bodyweight < 60 kg). Some patients initiated lenvatinib at a reduced dose owing to poor general condition. Dose reduction or discontinuation owing to AEs was decided by physicians.

Lenvatinib-induced hypothyroidism was treated according to the manuals published by the Japanese Ministry of Health, Labour and Welfare [33]. We monitored serum levels of TSH, free triiodothyronine, and free thyroxine at baseline and monthly thereafter. A patient was administered levothyroxine sodium at a dose of 25 µg when she/he had overt hypothyroidism with TSH values over 10 µg/mL. The dose of levothyroxine sodium was adjusted depending on the serum TSH values.

### 4.4. Clinical Evaluation

Treatment efficacy was evaluated according to the modified Response Evaluation Criteria in Solid Tumors [34] 2–3 months after the initiation of therapy. The DCR was defined as the percentage of patients with CR, PR, and SD. AEs were assessed using the National Cancer Institute Common Terminology ver. 4.03 [35] at month 3 for landmark analysis and during the follow-up period. Patients were followed up until 19 November 2019, or death.

### 4.5. Statistical Analysis

Cox regression models were used to detect contributing factors for treatment duration and OS. We used landmark analysis [36] to detect the relationship between clinical outcomes and AEs at month 3. Event analysis was performed using Kaplan-Meier method, and the statistical significance was determined using the log-rank test.

Next, Cox regression models were used to detect contributing factors for treatment duration and OS with respect to age-adjusted HR. To determine the effect of hypothyroidism on outcomes, HR and 95% CIs were calculated using Cox regression after adjusting potential confounding factors [37]. In addition, a sensitivity analysis was performed by altering the models. As entered variables of Models 1, 2, 3, and 4, age, hypothyroidism grade, sex, Child-Pugh score, hypoalbuminemia grade, TNM stage, and BCLC stage were selected based on previous reports on clinical importance to outcomes [19,21,24,38,39]. We consider that the number of variables that could be entered into the model was four at the maximum because of the small sample size. HRs for hypothyroidism grade were adjusted for age, BCLC, and sex in Model 1; age, BCLC, and hypoalbuminemia grade in Model 2; age, Child-Pugh, and TNM stage in Model 3; and age, TNM stage, and hypoalbuminemia grade in Model 4. Statistical analysis was performed using IBM SPSS version 26 (IBM Corp., Armonk, NY, USA). The significance level was set at *p <* 0.05.

## 5. Conclusions

This study suggested that better baseline liver function was predictive of longer treatment duration and better prognosis in patients with advanced HCC treated with lenvatinib. Moreover, an AE of Grade 2/3 hypothyroidism was associated with a better prognosis in patients receiving lenvatinib treatment for advanced HCC. Continuing anticancer therapy with appropriate thyroid hormone replacement may contribute to longer survival.

## Figures and Tables

**Figure 1 cancers-12-03078-f001:**
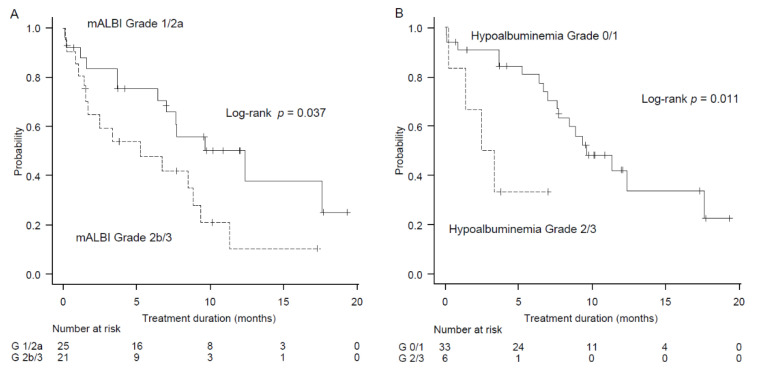
Factors associated with treatment duration. (**A**) The relationship between modified albumin bilirubin (mALBI) grade and treatment duration. (**B**) The relationship between hypoalbuminemia and treatment duration.

**Figure 2 cancers-12-03078-f002:**
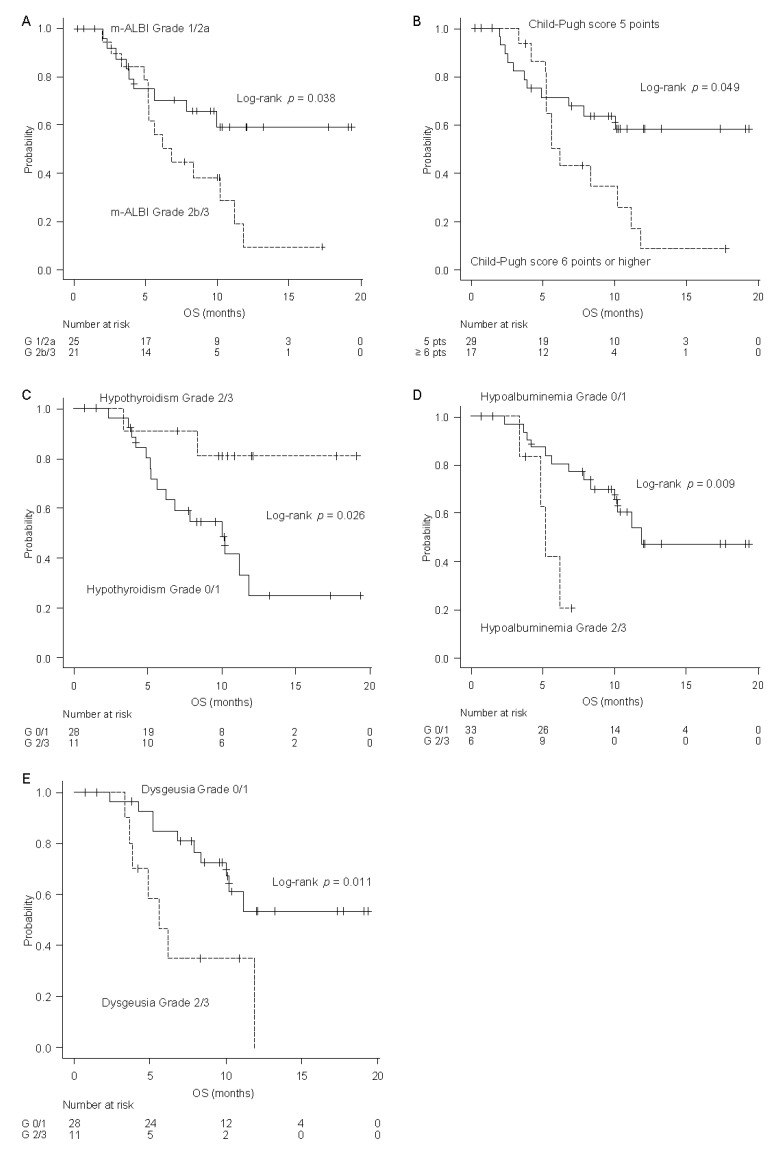
Factors associated with overall survival. (**A**) The relationship between modified albumin bilirubin (mALBI) grade and overall survival. (**B**) The relationship between Child-Pugh score and overall survival. (**C**) The relationship between hypothyroidism and overall survival. (**D**) The relationship between hypoalbuminemia and overall survival. (**E**) The relationship between dysgeusia and overall survival.

**Table 1 cancers-12-03078-t001:** Baseline demographic and clinical characteristics (*n* = 46).

Variable	Number of Cases (%)	
Sex	Male	37 (80)
	Female	9 (20)
Age, years	<75	24 (52)
	≥75	22 (48)
Body mass index	≥21	37 (80)
	<21	9 (20)
Etiology	HCV	14 (30)
	HBV	7 (15)
	Alcoholic	7 (15)
	NASH	4 (9)
	unknown	14 (30)
Child-Pugh score	5	29 (63)
	6	10 (22)
	≥7	7 (15)
mALBI grade	1	14 (30)
	2a	11 (24)
	2b	18 (39)
	3	3 (7)
TNM stage	III	18 (39)
	IV	28 (61)
BCLC (Kindai criteria)	B1	2 (4)
	B2	19 (41)
	C	25 (54)
Tumor size, mm	<50	27 (59)
	≥50	19 (41)
Extrahepatic invasion	Yes	20 (43)
Vascular invasion	Yes	9 (20)
Previous curative treatment	Yes	26 (56)
AFP, ng/mL	≥100	24 (52)
	<100	21 (48)
DCP, mAU/mL	≥1000	19 (41)
	<1000	27 (59)
Initial dose of lenvatinib	4 mg	18 (40)
	8 mg	15 (33)
	12 mg	13 (28)

HCV, hepatitis C virus; HBV, hepatitis B virus; NASH, non-alcoholic steatohepatitis; mALBI grade, modified albumin bilirubin grade; TNM, tumor-node-metastasis; BCLC, Barcelona clinic liver cancer; AFP, alpha-fetoprotein; DCP, des-gamma-carboxy prothrombin.

**Table 2 cancers-12-03078-t002:** Adverse events (*n* = 46).

Adverse Events	Any Grade	Grade 1	Grade 2	Grade 3
Any adverse events	46 (100)	6 (13)	19 (41)	21 (46)
Fatigue	39 (85)	23 (50)	9 (20)	7 (15)
Anorexia	31 (67)	8 (17)	16 (35)	7 (15)
Proteinuria	23 (50)	13 (28)	3 (7)	7 (15)
Weight loss	23 (50)	13 (28)	9 (20)	1 (2)
Hypoalbuminemia	21 (46)	15 (33)	6 (13)	0
Skin toxicity ^a^	20 (43)	10 (22)	6 (13)	4 (9)
Hypothyroidism	19 (41)	6 (13)	12 (26)	1 (2)
Hoarseness	18 (39)	16 (35)	2 (4)	0
Nausea & vomiting	17 (37)	13 (28)	2 (4)	2 (4)
Diarrhea	16 (35)	14 (30)	1 (2)	1 (2)
Abdominal pain	16 (35)	12 (26)	4 (9)	0
Dysgeusia	13 (28)	2 (4)	11 (24)	0
Hypertension	11 (24)	5 (11)	5 (11)	1 (2)
Alopecia	11 (24)	8 (17)	3 (7)	0

Data are presented as the number of patients (%) during the follow-up period. ^a^ Skin toxicity includes a hand/foot skin reaction and any kind of skin trouble.

**Table 3 cancers-12-03078-t003:** Baseline demographic and clinical variables and adverse events associated with treatment duration.

Variable	Age Adjusted HR (95% CI)	Multivariate ^a^				
		Model 1	Model 2	Model 3	Model 4	
		HR (95% CI)	HR (95% CI)	HR (95% CI)	HR (95% CI)	
Baseline characteristics						
Body mass index ≥ 21 (vs. < 21)	0.81 (0.21–1.23)					
HCV infection (vs. other etiology)	1.67 (0.77–3.65)					
Child-Pugh score = 5 (vs. ≥ 6)	0.46(0.21–1.00)			0.45 (0.19–1.08)		
mALBI = 1/2a (vs. other)	0.43 (0.21–0.99)					
TNM stage III (vs. IV)	1.27 (0.59–2.72)			1.00 (0.39–2.55)	1.30 (0.52–3.27)	
BCLC B1/2 (vs. C)	1.42 (0.66–3.04)	1.62 (0.63–4.18)	1.35 (0.55–3.30)			
Maximum tumor size < 50 mm (vs. ≥ 50 mm)	1.02 (0.48–2.16)					
Extrahepatic invasion − (vs. +)	1.79 (0.79–4.05)					
Vascular invasion − (vs. +)	0.55 (0.24–1.27)					
Previous curative therapy: Yes (vs. No)	1.76 (0.84–3.70)					
AFP < 100 (vs. ≥ 100)	0.64 (0.30–1.39)					
DCP < 1000 (vs. ≥ 1000)	0.56 (0.27–1.20)					
Reduced initial dose No (vs. Yes)	0.88 (0.41–1.93)					
Sex, Male (vs. Female)	0.87 (0.35–2.16)	0.87 (0.27–2.87)				
Adverse events ^b^						
Hypoalbuminemia grade 2/3 (vs. grade 0/1)	4.38 (1.28–15.00)		5.01 (1.37–18.35)		5.65 (1.55–20.53)	
Hypothyroidism grade 2/3 (vs. grade 0/1)	0.49 (0.18–1.36)	0.57 (0.18–1.74)	0.44 (0.16–1.23)	0.52 (0.17–1.56)	0.46 (0.16–1.33)	

HR, hazard ratio; CI, confidence interval; HCV, hepatitis C virus; TNM, tumor-node-metastasis; BCLC, Barcelona Clinic Liver Cancer; AFP, alpha-fetoprotein; DCP, des-gamma-carboxy prothrombin. ^a^ Univariate and multivariate analyses were performed using Cox proportional hazards regression analysis (*n* = 49). ^b^ Association of adverse events with treatment duration was evaluated with landmark analysis at month 3 (*n* =3 9). HRs for hypothyroidism grade were adjusted for age, BCLC, and sex in Model 1; for age, BCLC, and hypoalbuminemia grade in Model 2; for age, Child-Pugh score, and TNM stage in Model 3; and for age, TNM stage, and hypoalbuminemia grade in Model 4.

**Table 4 cancers-12-03078-t004:** Baseline demographic and clinical variables and adverse events associated with overall survival.

Variable	Age Adjusted HR (95% CI)	Multivariate ^a^			
		Model 1	Model 2	Model 3	Model 4
		HR (95% CI)	HR (95% CI)	HR (95% CI)	HR (95% CI)
Baseline characteristics					
Body mass index ≥ 21 (vs. < 21)	0.48 (0.18–1.27)				
HCV infection (vs. other etiology)	1.67 (0.70–3.96)				
Child-Pugh score = 5 (vs. ≥ 6)	0.44 (0.19–1.00)			0.30 (0.11–0.82)	
mALBI = 1&2a (vs. other)	0.44 (0.19–1.04)				
TNM stage III (vs. IV)	1.05 (0.45–2.46)			0.48(0.16–1.50)	0.59 (0.20–1.68)
BCLC B1/2 (vs. C)	1.24 (0.54–2.85)	1.00 (0.34–2.92)	0.76 (0.26–2.22)		
Tumor size < 50 mm (vs. ≥ 50 mm)	0.90 (0.40–2.06)				
Extrahepatic metastasis − (vs. +)	1.32 (0.56–3.13)				
Vascular invasion − (vs. +)	0.58 (0.24–1.40)				
Previous curative therapy: Yes (vs. No)	1.39 (0.61–3.16)				
AFP < 100 (vs. ≥ 100)	0.69 (0.30–1.61)				
DCP < 1000 (vs. ≥ 1000)	0.63 (0.28–1.43)				
Reduced initial dose No (vs. Yes)	0.55 (0.24–1.26)				
Sex, Male (vs. Female)	1.06 (0.37–2.99)	1.34 (0.36–4.90)			
Adverse events ^b^					
Dysgeusia grade 2/3 (vs. grade 0/1)	3.55 (1.27–9.88)				
Hypoalbuminemia grade 2/3 (vs. grade 0/1)	4.96 (1.36–18.12)		5.83 (1.45–23.48)		5.83 (1.54–22.10)
Hypothyroidism grade 2/3 (vs. grade 0/1)	0.21 (0.05–0.94)	0.20 (0.04–0.98)	0.19 (0.04–0.87)	0.18 (0.04–0.87)	0.16 (0.04–0.77)

HR, hazard ratio; CI, confidence interval; HCV, hepatitis C virus; TNM, tumor-node-metastasis; BCLC, Barcelona Clinic Liver Cancer; AFP, alpha-fetoprotein; DCP, des-gamma-carboxy prothrombin. ^a^ Univariate and multivariate analyses were performed using Cox proportional hazards regression analysis (*n* = 49). ^b^ Association of adverse events with treatment duration was evaluated with landmark analysis at month 3 (*n* = 39). HRs for hypothyroidism grade were adjusted for age, BCLC, and sex in Model 1; for age, BCLC, and hypoalbuminemia grade in Model 2; for age, Child-Pugh score, and TNM stage in model 3; and for age, TNM stage, and hypoalbuminemia grade in Model 4.

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
