# Peer review of "Hypothyroidism Is a Predictive Factor for Better Clinical Outcomes in Patients with Advanced Hepatocellular Carcinoma Undergoing Lenvatinib Therapy"

_cancers, 2020, doi:10.3390/cancers12113078_

Round 1
Reviewer 1 Report
This is a revised version of a previously submitted study. Although the authors have addressed some of the raised issues, I believe major limitations remain unresolved.
Reviewer 2 Report
The Authors have modified the manuscript according to my suggestions.
While the manuscript is now more sound, the Authors forgot to report the E-values of their sensitivty analyses for the hazard ratios. Please refer to https://doi.org/10.7326/M16-2607
If the 95% CI of the E-value is quite close to the null (as I imagine), more cautios conclusions are needed.
This manuscript is a resubmission of an earlier submission. The following is a list of the peer review reports and author responses from that submission.
Round 1
Reviewer 1 Report
Dr. Masako Shomura et al. reported that hypothyroidism is a predictive factor for better clinical outcomes in patients with advanced hepatocellular carcinoma (HCC) treated with lenvatinib using their small (less than 50) retrospective cohort. The relationship between one of the adverse events, hypothyroidism, and better prognosis is very interesting in a practical clinical setting. The relationship which the authors reported is correct. However, generally speaking, the analysis by the post-treatment events must have many biases, especially in the small cohort. Therefore, the interpretation and discussion of data and analysis should be paid much more attention.
Major
- The relationship between hypothyroidism during lenvatinib treatment and overall survival (OS) has been reported in the meeting (ASCO-GI 2019). In the meeting, Dr. Max Sung et al. reported the association between OS and adverse evets with lenvatinib treatment using data from the REFLECT study (n=478). In the report, they concluded that the occurrences of hypertension, diarrhea, proteinuria, or hypothyroidism were associated with more prolonged survival. The authors should refer and discuss the report, although the report was not published as an article.
- In table 1, the authors should show the Child-Pugh score 6 and ≥7, because the percentage of Child-Pugh class B is usually essential for data interpretation.
- Usually, treatment efficacy is calculated by the intension-to-treat population. The authors excluded two patients who died before the first evaluation. These two patients should be included in the calculation.
- The authors reported that two patients died before the first evaluation. They should describe the cause of death, disease progression, or adverse events.
- For analysis of factors associated with treatment duration, the authors should show the basement data,
- The authors should explain their criteria for hypothyroidism treatment. In a practical setting, it is complicated to distinguish the general malaise from lenvatinib treatment or that from hypothyroidism. So, the authors set some criteria, for example, replace treatment start in TSH ≥10 µIU/mL or not. The criteria are critical because it effects on the grading of hypothyroidism.
- The treatment duration was usually associated with adverse events. The more prolonged treatment usually causes more adverse events. Therefore, the authors should show the treatment duration difference in hypothyroidism grades.
- The authors analyzed a total of 46 patients in this study. However, in factorial analysis with OS, the number of patients was decreased to 39 patients in the analysis of adverse events. Seven patients (15%) have been missed. The exclusion is an unusual analysis, and it must cause arbitrary bias.
- In the discussion, the authors discussed patients who progress to hypothyroidism might be more sensitive to TKI than those who do not. For this discussion, the authors should check the response in their cohort.
Minor
- In figure 2, “0” and “1” in patients at risk should be changed. “Log rank” should be “Log-rank.”
Reviewer 2 Report
The present study aims to identify predictive factors of treatment duration and overall survival (OS) in patients with advance HCC undergoing lenvatinib therapy. The authors included 46 patients (enrolled over a period of 19 months); however treatment efficacy was evaluated only in 44 patients because two patients died before the first evaluation. Median treatment duration was 7.7 months and OS 10.2 months
Similarly to previous studies, preserved hepatic function (albumin, Child) was a favorable predictive factor for both treatment duration and OS. Surprisingly, severe hypothyroidism was also associated to a better OS.
As the authors themselves agree upon, the study presents major limitations: this a small monocentric study. confounding between variables was unavoidable. Further, the same findings have been already described in small cohorts as the one presented here. The cohort size do not have enough power to prove nor contrast previous reports
Reviewer 3 Report
I read with pleasure this paper about the relationship between hypothyroidism and survival in patients with HCC treated with lenvatinib. The topic is of interest and the rationale behind this study is good, as some AEs related to TKIs have a prognostic significance. There are, however, some important concerns.
MAJOR
- The population size is relevant for a single center, but still too small for some survival analysis. This impression is conformed when reading the 95% confidence intervals of the hazard ratio, which are often very large and thus little informative. Moreover, the Authors put a lot a varaibles in the multivariable analysis with a high risk of colinearity. As a rule of thumb, a maximum of a variable every 10 patients should be considered in the multivariable analysis to avoid his problem (and thus a maximum of 4 in this specific case. Should the Authors perform a sensitivity analysis to evaluate of strong a possible confounder to falsify their results (which I strongly advise to perform) they will probably find that even small confounders could nullify the statistical significativity. Probably, one or two patients with a different hypothydism status could disprove these results.
- The Authors correctly report other studies evaluating the relationship hypothyroidism-response for lenvatinib in HCC. These studies had been performed with more robust methods, so, unfortunately, this paper provide very limited novelty for a journal such as Cancers.
MINOR
- I am aware of the difficulties in reaching an higher number of patients in monocenter or bicenter studies. To increase the statistical power in such conditions, I suggest to replace the landmark analysisis statistics (which imply a loss of patients) with a time-dependent Cox regression considering the time of onset of hypothyroidism. Such analysis can be made without excluding patient which died before the landmark.